# The Impact of a Cervical Collar on Intracranial Pressure in Traumatic Brain Injury Patients: A Systematic Review and Meta-Analysis

**Naif Bazaie [1], Ibrahim Alghamdi [1], Naif Alqurashi [2] and Zubair Ahmed [1,3,4,*]**

[1]   Institute of Inflammation and Ageing, College of Medical and Dental Sciences, University of Birmingham, Edgbaston, Birmingham B15 2TT, UK; abu_turki789@hotmail.com (N.B.); Ibraghamdi@kku.edu.sa (I.A.)

[2]   Department of Accidents and Trauma, Prince Sultan Bin Abdulaziz College for Emergency Medical Services, King Saud University, Riyadh 11451, Saudi Arabia; Naif012n@gmail.com

[3]   Centre for Trauma Sciences Research, University of Birmingham, Edgbaston, Birmingham B15 2TT, UK

[4]   Surgical Reconstruction and Microbiology Research Centre, National Institute for Health Research, Queen Elizabeth Hospital, Birmingham B15 2TH, UK

*   Correspondence: z.ahmed.1@bham.ac.uk

**Abstract:** Purpose: Although the use of a cervical collar in the prehospital setting is recommended to prevent secondary spinal cord injuries and ensure spinal immobilization, it is not known what effects this has on raising intracranial pressure (ICP) in traumatic brain injury (TBI) patients. In the absence of studies measuring ICP in the prehospital setting, the aim of this study was to systematically review the data related to ICP changes measured after presentation at the hospital in patients who had arrived wearing cervical collars. Methods: We searched Medline (PubMed), Embase, CINAHL, and Google Scholar for studies that investigated in-hospital ICP changes in TBI patients arriving at the hospital wearing collars. Titles, abstracts, and full texts were then searched for inclusion in the study. A narrative synthesis, as well as a meta-analysis, was performed. Results: Of the 1006 studies identified, only three met the inclusion/exclusion criteria. The quality of the three included studies was moderate and the risk of bias was low. All three studies used the Laerdal Stifneck collar, but all studies showed an increase in ICP after application of the collar. A further three studies that measured ICP but did not fit the systematic search were also included due to low patient numbers. A meta-analysis of the pooled data confirmed a significant increase in ICP, although between the four studies, only 77 patients were included. The meta-analysis also confirmed that after removal of the collar, there was a significant decrease in ICP. Conclusions: Our study suggests that the use of a cervical collar increases ICP in TBI and head injury patients, which may have detrimental effects. However, due to the extremely low sample size from all six studies, caution must be exercised when interpreting these data. Thus, further high-quality research is necessary to unequivocally clarify whether cervical collars should be used in patients with TBI.

**Keywords:** traumatic brain injury; cervical collars; intracranial pressure; prehospital

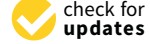



## 1. Introduction

Traumatic brain injury (TBI) is an increasing public health issue and contributes to trauma-associated injuries globally [1]. The most common causes of TBI are falls and road traffic accidents, which account for >50% of the total number of cases [2]. According to the World Health Organization, TBI is the leading cause of nearly half of all disabilities. The 2010 Global Burden of Disease project estimated that the annual global prevalence of TBI is approximately 15 million [3]. The incidence of TBI that results in hospital admission is estimated to be 262 cases per 100,000 individuals, in an evaluation of 16 European countries. In the United States, >1 million people incur TBIs, the prevalence of which is higher in males than in females. Studies have highlighted the negative outcome of TBI such as disability and memory impairments, which worsen with more severe TBI. In some cases,

patients are at risk of dementia [2]. In the previous decade, trauma programmes have highlighted that adequate prehospital management has successfully led to a decline in morbidity and mortality from TBI [4].

Prehospital care of patients with TBI is vital to the quality of hospital care, as it can significantly affect subsequent care outcomes. The aim of prehospital care is to minimize secondary injuries while optimizing the patient's well-being [5]. It also promotes health-saving emergency care in patients with TBI [6]. Cervical collars are most commonly used in patients at risk of cervical spine injury. Use of a cervical collar and establishment of control airways are the first measures applied in a prehospital setting to improve status and prevent secondary injury to the spinal cord. Cervical collars have been used for >30 years and are considered essential in modern prehospital trauma care. Patients with TBI are at a high risk of spinal injury and thus require a rigid cervical collar to promote spinal stability [7,8]. In traumatic injuries, rigid cervical collars have been widely considered as immobilizers because they offer superior cervical restrictions [9]. An example of a cervical collar utilized by emergency medical services (EMS) is the Laerdal Stifneck [10]. Globally, most ambulance services use the Laerdal Stifneck collar (Life-Assist, Inc., Rancho Cordova, CA, USA), which is one of the most universally used collars for this purpose in a prehospital setting [11–13]. However, spinal collars must be fitted properly and adjusted for height, level of fitness and should not interfere with the blood flow into or from the brain [13].

The increasing incidence of cervical collar-associated complications in patients with TBI is a cause of growing concern. For instance, cervical collars might alter the intracranial pressure (ICP) by compressing the jugular vein. In some cases, they have been associated with an increase in the prevalence rates of morbidity and mortality. Moreover, evidence shows that airway compromise can result from the inaccurate application of cervical collars [14]. In some cases, cervical collars prevent adequate evaluation of penetrating head wounds for evolving injuries, tracheal deviation, tissue oedema, emphysema, and hematoma, which can only be identified after removal of the cervical collar [15]. The above-mentioned complications can further result in neurological deterioration.

The aim of this systematic review is to comprehensively assess the research surrounding the impact of using cervical collars and its effects on ICP in patients with TBI. This is a potentially significant issue in patients with head injuries; therefore, the decision on whether to use a cervical collar before arrival at the hospital should be evidence based.

## 2. Materials and Methods

### 2.1. Literature Search

Two authors (N.B. and Z.A.) conducted a literature search following the guidelines in the Preferred Reporting Items for Systematic Reviews and Meta-Analysis (PRISMA) statement and the Cochrane Handbook for Systematic Reviews for Interventions to promote literature efficiency and relevance [16] The title and keywords were searched with the following words singly and in combination: "traumatic brain injury", "cervical collar", "intracranial pressure", and "prehospital". We searched Medline (PubMed) via Ovid, Embase via Ovid, CINAHL, and Google Scholar. Google Scholar was searched for keywords in various studies using the forward and backward strategies. We also screened the reference list of the articles to identify possible studies for inclusion.

### 2.2. Inclusion and Exclusion Criteria

An effective population, intervention, control, and outcomes (PICO) question relevant to the inclusion criteria was initially formulated [17]. Owing to the fact that the systematic review was focused on patients with TBI, the following PICO components were formulated: adult patients aged >16 years with a TBI caused by trauma, with cervical collar immobilization before arrival at the hospital (prehospital setting), and evaluation of changes in ICP after cervical collar application as the primary outcome. Other inclusion criteria were as follows: (1) research articles published in English, original and peer reviewed; (2) randomized controlled trials and prospective, retrospective, and cohort studies that

focused primarily on the use of cervical collars applied in the prehospital setting; (3) in the case of more than one publication emanating from a single study, the version with the most extended follow-up and full reporting was included in the present study.

We excluded review articles, case reports and conference abstracts, animal-based studies, lab or computer simulations, articles published before 1 January, 1990, and studies with long-term care/rehabilitation (Table 1).

**Table 1.** Inclusion and exclusion criteria for systematic review.

| Inclusion Criteria | Exclusion Criteria |
|---|---|
| <ul><li>Research articles published in English.</li><li>All patients with TBI.</li><li>Randomized control trials, prospective, retrospective, and cohort studies focusing primarily on the use of cervical collars applied in the prehospital setup.</li><li>Adults (>16 y/o).</li></ul> | <ul><li>Review articles, case reports, volunteer's studies, and conference abstracts.</li><li>Article populated before 1 January 1990.</li><li>Studies with long-term care/rehabilitation.</li><li>Animal-based research.</li></ul> |

*2.3. Data Collection Process*

Two independent reviewers (N.B. and Z.A.) conducted the literature search based on the keywords. Titles and abstracts were searched and articles identified for full-text review. Cases of disagreements were resolved by discussion. Screened articles had to meet the inclusion/exclusion eligibility criteria, records were exported into RAYYAN (Intelligent Systematic Review; https://www.rayyan.ai/, last accessed 20 November 2021), duplicate studies were excluded, titles and abstracts were inspected manually, and articles were retrieved [18]. Where full texts were unavailable, we wrote to the authors, but none responded in time for inclusion in this review.

*2.4. Data Extraction and Synthesis*

The data extraction process was completed by the N.B and checked for accuracy by Z.A. A customized data extraction form was created using Microsoft Excel. Each study was read with the aim of extracting the following data: (1) study characteristics (titles and study designs); (2) patient characteristics (number of patients and ages); (3) aims of study and limitations; and (4) findings relevant to the use of cervical collars.

*2.5. Quality Assessment and Statistical Analysis*

The Newcastle–Ottawa Scale (NOS) was used to assess the quality and risk of bias of the nonrandomized studies [19]; each assessment criteria are assigned a maximum of four stars (selection domain), two stars (comparability domain), and three stars (outcome domain). The maximum possible number of stars is nine, and studies with less than five stars have a high risk of bias. This approach assessed the quality and the risk of bias for the included studies. The risk for bias was evaluated using Egger's test; a value of $p < 0.1$ was regarded as statistically significant. Despite the differences in methods used to record ICP and the time of collar application, we performed a meta-analysis using Review Manager 5.4.1 (Cochrane Informatics & Technology, London, UK) using the random effects model.

**3. Results**

*3.1. Study Selection*

The initial systematic search using the PRISMA strategy [16] yielded 1006 studies from Medline (PubMed), Embase, CINAHL, and Google Scholar that met the inclusion criteria (Figure 1). From these, we excluded 154 non-English written studies, 210 that were animal-based articles, 198 studies that included children aged less than 16 years old, and 121 studies that did not have full text. Additionally, we excluded 200 studies that were published before 1 January, 1990, as well as 66 that were irrelevant to the topic. After

full-text reading, 54 studies were excluded, as they were not relevant to the fundamental subject area. Of these studies, three could be included in our systematic review.

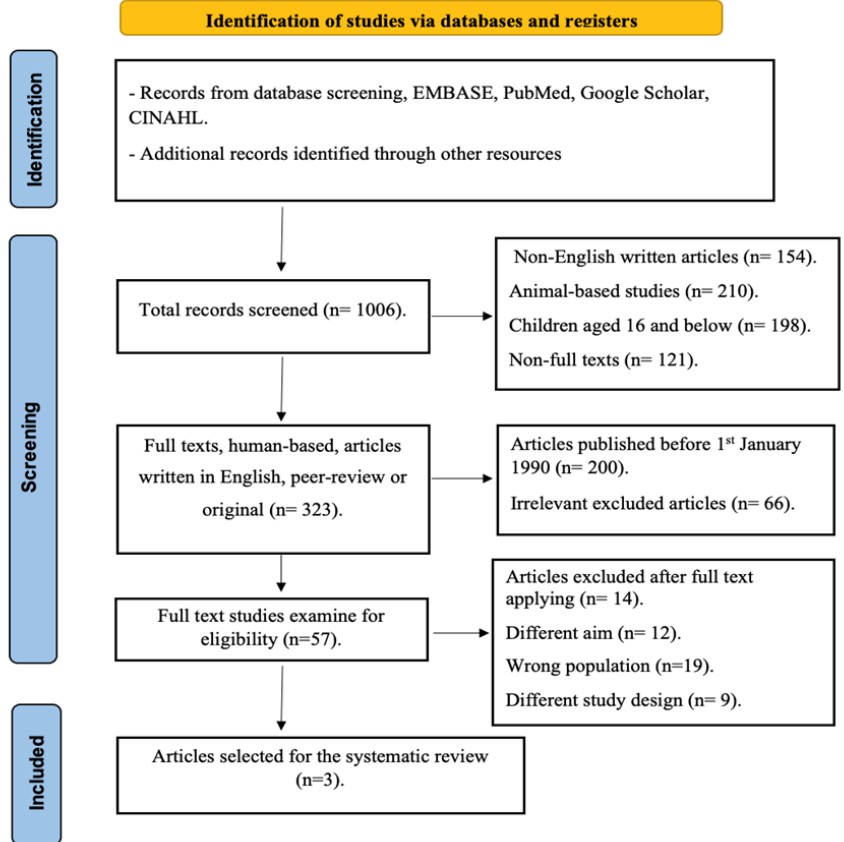

**Figure 1.** PRISMA flowchart of the screening process.

*3.2. Study Characteristics*

Table 2 shows the characteristics of the three included studies identified by the systematic search. In all of the studies, a prospective observational study design was used as the study methodology. The studies included in this review were conducted in two areas: United Kingdom (*n* = 2) [20,21] and Australia (*n* = 1) [10]. Fifty-nine patients were included overall, with the same number of patients in each group in each study. All of the studies employed the Laerdal Stifneck collar. One study did not report % of males [20], but in the remaining two studies, males constituted 60 [21] and 80% [10] of the total number of patients, respectively. All ICP measurements were performed within 48 h of presentation at the hospital using the same collar that patients arriving at the hospital were wearing. Although the included studies used different ICP measurement devices (Table 2), they were all invasive recording methods that included use of an external ventricular drain, a microventricular catheter, and a Codman microsensor [10,20,21].

**Table 2.** Characteristics of the included studies identified from the systematic search.

| Authors | Year | Study Design | Country | Sample Size | Intervention | GRADE Quality | ICP Measurement Device | Findings |
|---|---|---|---|---|---|---|---|---|
| Mobbs et al. [10] | 2002 | A prospective, observational study | Australia | 10 | Laerdal Stifneck collars | Moderate | Medtronic, external ventricular drain, Camino (invasive) | Increase in ICP (mean rise of 4.4 mmHg) following application of the collar. |
| Davies et al. [20] | 1996 | A prospective, observational study | UK | 19 | Laerdal Stifneck collars | Moderate | Camino microventricular catheter (invasive) | Increase in ICP (mean rise of 4.5 mmHg) following application of the collar. |
| Hunt et al. [21] | 2001 | A prospective, observational study | UK | 30 | Laerdal Stifneck collars | Moderate | Codman microsensor (invasive) | Increase in ICP (mean rise of 4.6 mmHg) following application of the collar. |

Since so few patients were reported in the three studies from our systematic review, we also included three other studies to the narrative synthesis that were outside our inclusion/exclusion criteria but were pertinent to the subject area of spinal collars and ICP after head injury (Table 3). These included one study that was written in German, containing 18 patients that received either the Speith or Philadelphia cervical collars and had an invasive epidural transducer to record ICP [22]. Another study in the UK had nine patients with a cervical collar that was not identified, and the ICP measurement device was also not stated [23]. Little information regarding this publication could be found except a summary of the ICP data before and during the application of the collar, and hence it was not possible to assess the quality of the article. Finally, the third study was a case report study from two patients in the UK that used the Laerdal Stifneck cervical collar and measured ICP using an invasive monitor (unidentified) [24].

**Table 3.** Characteristics of the additional studies identified from the systematic search but were outside our inclusion/exclusion criteria.

| Authors | Year | Study Design | Country | Sample Size | Intervention | GRADE Quality | ICP measurement Device | Findings |
|---|---|---|---|---|---|---|---|---|
| Kuhnigk et al. [22] | 1993 | A prospective observational study | Germany | 18 | Speith (n = 12) and Philadelphia cervical collars (n = 6) | Moderate | Epidural transducer (invasive) | No significant change in ICP. |
| Porter et al. [23] | 1993 | A prospective observational study | UK | 9 | * Cervical collar | Not assessed | N/A | Increase in ICP (mean rise of 9.9 mmHg) following application of the collar. |
| Craig and Nielsen [24] | 1991 | Case report | UK | 2 | Laerdal Stifnek collars | Low | ICP monitor (invasive) § | Increase in ICP (18 mmHg and 15 mmHg) following application of the collar. |

Notes: N/A = information not available; ICP = intracranial pressure; * = no information available about the type of cervical collar used; § No other information given.

In all of the studies except one [22], there was a significant increase in ICP during the application of the collar, ranging from 4.4 to 18 mmHg (Tables 2–5) [10,20,21,23,24]. In two

studies, this rise in ICP during the application of the collar was reduced to near normal, before collar application levels after the removal of the collar [20,21]. In one study, ICP after removal of the collar was not reported [10]. The duration of collar application also varied greatly and ranged from 3 to 20 min (Table 4). In two studies, the rise in ICP after cervical collar application was 9.9–18 mmHg, one of which only had nine patients, and the other were case reports from two study participants [23,24]. These studies reported the highest levels of ICP rises, but unfortunately, the quality of one could not be assessed due to the full paper being unavailable [23], whilst the other was deemed low quality, as it was from case reports [24].

**Table 4.** ICP measurements (mean $\pm$ SD/mmHg) before, during, and after the application of cervical collars from studies identified by the systematic search.

| Study | Before Application | During Application | After Removal | Time of Collar Application (min) |
|---|---|---|---|---|
| Mobbs et al. [10] | 20.5 $\pm$ 14.2 | 25.8 $\pm$ 11.5 | NR | 3–5 |
| Davies et al. [20] | 13.7 $\pm$ 5.7 | 18.3 $\pm$ 7.3 | 14.4 $\pm$ 6.0 | 20 |
| Hunt et al. [21] | 14.1 $\pm$ 6.6 | 18.8 $\pm$ 8.4 | 14.3 $\pm$ 6.6 | 5 |

**Table 5.** ICP measurements (mean $\pm$ SD/mmHg) before, during, and after the application of cervical collars from additional studies.

| Study | Before Application | During Application | After Removal | Time of Collar Application (min) |
|---|---|---|---|---|
| Kuhnigk et al. [22] | 17.0 $\pm$ 6.1 | 17.7 $\pm$ 6.7 | 17.2 $\pm$ 5.9 | 10 |
| Porter et al. [23] | 12.8 (range 6–19) | 22.7 (range 24–36) | NR | 5 * |
| Craig and Nielson [24] | 14 | 32 | 14 | 15 |
| | 10 | 25 | 10 | 12 |

Notes: NR = not reported; min = minutes; * collars were applied and left in place until ICP was steady for 5 min before measuring.

### 3.3. Risk of Bias

In general, the risk of bias was low for the observational studies identified by our systematic search using NOS [19]. All studies were given one star for the ascertainment of intervention, while the stars of outcomes were presented at the beginning of each study for two studies [10,20], the stars of the representativeness of the intervention cohort for two studies [20,21] from the categories of the selection bias domain. For comparability, all studies were assigned two stars for each place. In the outcome, three stars were selected for this part because all of the studies reported assessment of outcomes domain, timing, and adequacy of follow-up (Table 6).

**Table 6.** Quality assessment using the Newcastle–Ottawa scale (NOS) of the included studies identified from the systematic search. Green = high quality; yellow = moderate quality.

| | Newcastle–Ottawa Scale Results | | | |
|---|---|---|---|---|
| Reference | Selection (4) | Comparability (2) | Outcome (3) | Total (9) |
| Mobbs et al. [10] | 2 | 2 | 3 | 7 |
| Davies et al. [20] | 3 | 2 | 3 | 8 |
| Hunt et al. [21] | 2 | 2 | 3 | 7 |

In the studies included, of those identified by our systematic search, one study was graded as moderate [22], one study was graded as low [24], whilst the other study did not have full text available to be able to determine the quality [23] (Table 3).

### 3.4. Meta-Analysis for ICP during and after Collar Application

Despite the differences in the time of collar application after arrival at the hospital, we performed a meta-analysis with the pooled data from the 77 patients in the four

studies [10,20–22] identified from our systematic review to calculate the overall change in ICP during and after cervical collar application. The increase in ICP during collar application was statistically significant, with an overall mean increase of 3.57 mmHg (95%CI = 1.35, 5.79; *p* = 0.002; $I^2$ = 0%) (Figure 2). In contrast, after removal of the collar, reported in three of the four studies [20–22], there was a statistically significant mean decrease in ICP of 3.1 mmHg (95%CI = −5.52, −0.68; *p* = 0.01; $I^2$ = 12%) (Figure 3).

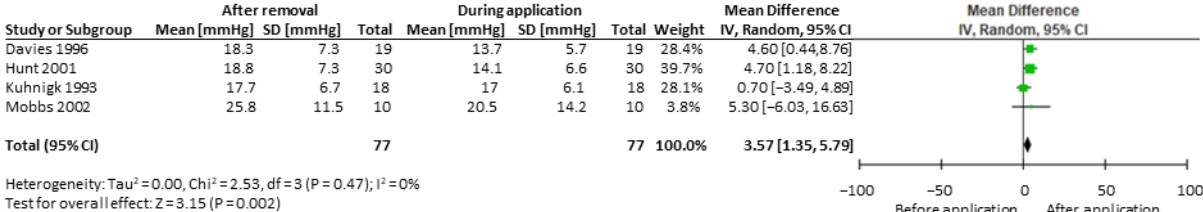

**Figure 2.** Forest plot of ICP changes during application of cervical collars in TBI patients identified by the systematic search.

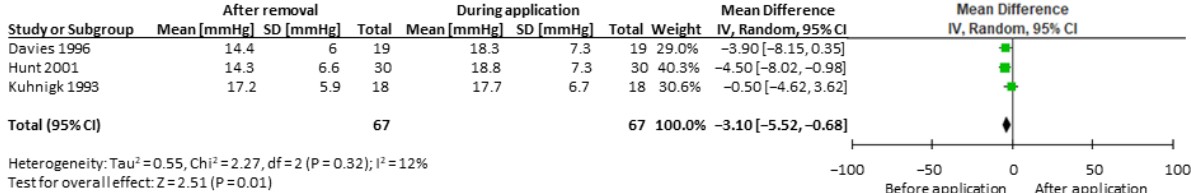

**Figure 3.** Forest plot of ICP changes after removal of cervical collars in TBI patients identified by the systematic search.

## 4. Discussion

This systematic review examined the currently available literature relevant to the use of cervical collars in the management of TBI. Three articles met the inclusion/exclusion criteria and were analyzed in this systematic review, whilst three other studies, which also measured ICP but did not meet all of our inclusion/exclusion criteria, were also added due to small sample sizes. All of the studies were prospective studies with the primary common theme that TBI patients arrived at the hospital wearing collars. During the testing period, reapplication of the collar led to a statistically significant increase in ICP. Our meta-analysis also confirmed that after collar removal, there was a statistically significant reduction in ICP. Therefore, our findings indicate that cervical collar application significantly raises the ICP in TBI patients, and hence clinicians should aim for early removal of collars in TBI patients in favor of alternative immobilization methods. Owing to the lack of prehospital data, a more extensive study must be conducted to clarify the protocols for use of cervical collars in patients with TBI in the prehospital setting.

Although spinal immobilization is commonly used for suspected spinal injuries, there is no solid evidence to support its use. There is even less evidence that spinal collars are beneficial, and many recommend that the use of cervical collars be minimized to prevent secondary injuries in TBI patients [25,26]. In support of this, several studies have shown that ICP increases due to the application of cervical collars [27–30]. Other issues such as discomfort, pressure ulcers, and jugular vein occlusion have been suggested as complications of cervical collars [31]. Intracranial hypertension is also associated with increased mortality and worsened neurological outcomes after TBI [24]. Therefore, the use of spinal collars and their benefits remain controversial.

Our systematic review suggests that the use of a rigid cervical collar was associated with a small but significant rise in ICP during its use [10,20–24]. In two of the three studies that measured ICP after removal of the collar, a significant decrease to precollar application levels was observed [20,21]. These studies demonstrate that ICP rises during cervical collar application and, in general, returns to near normal precollar application levels once

the collar is removed. These findings may be explained by a variety of factors that can affect ICP. For example, jugular vein compression is a main cause of ICP rises, and in this study, it is possible that the use of the rigid collar affected the magnitude of jugular vein compression [20,32,33]. Another proposed mechanism is the obstruction of the venous flow caused by a rise in internal jugular pressure, hindering intracranial venous flow due to the pressure on the jugular veins. Rigid cervical collars lead to compression and the deformation of the neck and hence hinder venous drainage [21]. The overall rise in ICP is complex and multifactorial, and other contributors such as impaired metabolism, tissue hypoxia, and mitochondrial dysfunctions also need to be considered [33]. Although there is no consensus as yet regarding the cut-off for ICP values, which are likely to lead to poorer outcomes in TBI, existing research suggests that treatment for increased ICP should begin when the pressure reaches 20–25 mmHg [21]. Such treatments may include eliminating physical factors that contribute to adverse outcomes [21].

Mobbs et al. [10] suggested dividing patients into three groups according to their baseline ICP when they present to the hospital. The patients in groups A and B had poor outcomes after collar application but showed no significant differences in ICP due to the mechanism of injury and continuous care needed. By contrast, group C had a positive outcome in terms of ICP after the early removal of the collar and cervical spine assessment. Both Mobbs et al. [10] and Hunt et al. [21] suggest that the early removal of hard cervical collars in traumatic head injury patients will potentially reduce ICP in some patients. Davies and colleagues [20] described two outcomes following the use of rigid cervical collars; in some patients, no change in ICP was observed, whereas in some patients, ICP increased after rigid cervical collars were used. The authors suggested that the effects of using rigid cervical collars on the ICP are associated with the patient's position on the intracranial compliance curve and individual anatomical differences. Thus, EMS staff should strive to remove cervical collars as early as possible and perform adequate cervical spine evaluation in head trauma patients before discontinuation of temporary stabilization. Until further studies have been conducted to fully understand the mechanism of cervical collars and the apparent increase in ICP, it is important for a radiological and medical evaluation of the cervical spine as soon as the patient arrives at the hospital.

*Limitations*

This systematic review has many limitations. For example, there are no studies where ICP is measured after the application of cervical collars in the prehospital setting, and hence there is a lack of direct evidence in this cohort of patients. Only three studies met the inclusion/exclusion criteria, with a total sample size of 59. This is too few studies, and the sample size is also too small to reach valid conclusions. In addition, the small sample sizes lead to caution in interpreting the statistical analysis presented in this study, which is further weakened by the different methods of measuring ICP. All of the included studies followed a retrospective study design; however, a retrospective design is highly discouraged owing to the challenges in obtaining ethical permission, lack of consent from patients, and possible outcomes without understanding the full effects [34]. The collars were in place for a very short time during the recording period (3–20 min) as opposed to the prehospital setting where collars may be in place for hours. Hence, any changes observed during the recording period could have been transient. Although this systematic review followed the PRISMA guidelines, various other limitations should be considered. To begin with, we excluded articles written in a non-English language, which meant that some articles may have been missed. In addition, the unavailability of 120 articles for full-text reading, despite us writing to the authors, poses a significant limitation, and we may have missed articles as a result.

**5. Conclusions**

Most EMSs utilize cervical collars for patients with TBI. The studies evaluated in this systematic review suggested that the use of cervical collars increased ICP in TBI patients.

Although the rises in ICP were small but significant, the impact of this small rise in ICP is currently unknown. This small rise may affect venous drainage, increase the prevalence of morbidity and mortality, and compromise the airways, which are among the factors that prehospital care primarily aims to prevent. However, the results of this study are to be interpreted with caution because it is based on a small number of studies with a small number of patients. Thus, our systematic review highlights the need for additional high-quality research on the use of cervical collars for patients with TBI.

**Author Contributions:** Conceptualization, N.B., Z.A.; methodology, N.B., I.A., N.A.; formal analysis, N.B., I.A., N.A., Z.A.; investigation, N.B., I.A., N.A.; data curation, N.B., I.A., N.A.; writing—original draft preparation, N.A.; writing—review and editing, N.A., I.A., N.B., Z.A.; supervision, Z.A.; project administration, N.B. All authors have read and agreed to the published version of the manuscript.

**Funding:** This research received no external funding.

**Institutional Review Board Statement:** Ethical review and approval were waived for this study as per advice from the NHS Health Research Authority (UK) decision tool, since it is a systematic review of published literature.

**Informed Consent Statement:** Patient consent was waived because no patients or members of the public were involved in the design, conduct of this study, or reporting of this research.

**Data Availability Statement:** All data generated as part of this study are included in the article.

**Conflicts of Interest:** The authors declare no conflict of interest.

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
