# Peer review of "The Impact of a Cervical Collar on Intracranial Pressure in Traumatic Brain Injury Patients: A Systematic Review and Meta-Analysis"

_traumacare, doi:10.3390/traumacare2010001_

Round 1

Reviewer 1 Report

I would like to congratulate the authors on their work. It is of very high quality but I do have a few comments/queries.

The title is a little misleading as this paper is on the effect of collars on ICP rather than TBI (although these are very closely linked). I would consider changing it. 

A structured abstract with headings will help readers immensely. The vast majority of readership is the abstract and in its current form leaves a bit to be desired.

I am concerned that some 120 papers were not able to be red in full text: especially in the context of such few included papers….

I understand the implication of time on trauma care, however I’m not sure if the time limitation in your case makes sense as it’s a very simple question of the impact of rigid collars on ICP; and to my knowledge ICP monitoring in the 90’s was as good as the 2000’s…. Nevertheless, I wonder if perhaps Kuhnigk could be included in the meta-analysis to help bolster the numbers as I can’t see a reason to exclude based time alone? If this is done, then 3 manuscripts will be able to be analysed on the effect of collar removal; as a question I have relates to if the raise in ICP is due to the natural history of TBI or due to the collar alone. It stands to reason that a reversible elevation in ICP would be far more likely to be due to the collar alone than natural history and therefore support your argument.

The quality of figure 2 is not at a level that can be published it seems.

I suggest shortening the conclusion as it is too long as it stands. Similarly, the discussion is rather cumbersome and could benefit from a haircut.

Line 321: May not be

Author Response

Comment: I would like to congratulate the authors on their work. It is of very high quality but I do have a few comments/queries.

Author response: Many thanks for commending our efforts.

Comment: The title is a little misleading as this paper is on the effect of collars on ICP rather than TBI (although these are very closely linked). I would consider changing it. 

Author response: We have amended the title to make it more appropriate.

Comment: A structured abstract with headings will help readers immensely. The vast majority of readership is the abstract and in its current form leaves a bit to be desired.

Author response: We have now structured the Abstract as recommended by the reviewer.

Comment: I am concerned that some 120 papers were not able to be red in full text: especially in the context of such few included papers….

Author response: There is no need to be concerned as the majority of these were abstracts for conference proceedings. With regard to full texts that were unavailable, these were >10 years old and we wrote to authors for full text but none of the authors responded during the time frame of this systematic review.  

Comment: I understand the implication of time on trauma care, however I’m not sure if the time limitation in your case makes sense as it’s a very simple question of the impact of rigid collars on ICP; and to my knowledge ICP monitoring in the 90’s was as good as the 2000’s…. Nevertheless, I wonder if perhaps Kuhnigk could be included in the meta-analysis to help bolster the numbers as I can’t see a reason to exclude based time alone? If this is done, then 3 manuscripts will be able to be analysed on the effect of collar removal; as a question I have relates to if the raise in ICP is due to the natural history of TBI or due to the collar alone. It stands to reason that a reversible elevation in ICP would be far more likely to be due to the collar alone than natural history and therefore support your argument.

Author response: This is a great suggestion and we have now included Kuhnigk et al into the meta-analysis and these are now presented as Figures 2 (old figure 3) and 3. We are now able to present data regarding before and after application and compare the data.  

Comment: The quality of figure 2 is not at a level that can be published it seems.

Author response: Figure 2 has been replaced with a table (now table 6) to show the same information.

Comment: I suggest shortening the conclusion as it is too long as it stands. Similarly, the discussion is rather cumbersome and could benefit from a haircut.

Author response: The discussion and the conclusions has been shortened.

Comment: Line 321: May not be

Author response: Amended (now Line 434).

Reviewer 2 Report

I commend the authors for the significant amount of work they have put in to this study. It has utilised a number of tools and guidelines to support a robust and methodologically sound systematic review.

I feel like I get what they were trying to do – illustrate that in addition to little evidence of benefit, collars may cause measurable harm in this particular cohort of patient.

Unfortunately from my perspective I felt like the study was confused about its objective. In particular it was not clear whether it was addressing a pre-hospital population, or a severe TBI/ICU population (in which you can measure ICP)

In the PICO question

-The population includes patients without a cervical collar?

-Primary outcome is “the effects of placement of the cervical collar in the patient with TBI” – this is very non-specific and then is not reported on anyway. Even if it was, I imagine you would need extremely large numbers to identify any associations between collar use and secondary TBI severity

-Secondary outcome is “changes in ICP after cervical collar placement”. By definition this can only be measured with an invasive device, so by definition these are post-op/ICU patients and the impact on ICP in the pre-hospital setting remains unknown. Additionally this is not a patient centred outcome – yes it’s true that poorly controlled ICP correlates with poorer functional outcomes but is a difference of 4.5mmHg relevant?

There are a number of unsupported statements in the conclusion

- “ICP is a contraindication of cervical collars” line 264 

- “Early removal and evaluation ensured minimal impact on the patient outcome.” Line 279

- Line 278 says “Emergency Medical Service should strive to remove cervical collars early” but then line 282 seems to support a radiological evaluation which can’t happen pre-hospital.

Line 308 All of the included studies were also limited to a single type of rigid cervical collar, the Stifneck collar, 309 which is widely used in the UK. – but then you have commented on the german study which used alternative collars.

You have not kept to your own inclusion criteria by including data from studies of unassessable quality, foreign language and inadequate study design.

At best, I would re-work this as a study demonstrating the small amount of collated data suggesting collars do increase ICP, and highlighting to providers that it may be worthwhile utilising alternative methods of spinal immobilisation (such as tape and sandbags) early in patients with TBI.

There are typos in line 249 and 252 "sue" not "use"

Author Response

Comment: I commend the authors for the significant amount of work they have put in to this study. It has utilised a number of tools and guidelines to support a robust and methodologically sound systematic review. I feel like I get what they were trying to do – illustrate that in addition to little evidence of benefit, collars may cause measurable harm in this particular cohort of patient. Unfortunately, from my perspective I felt like the study was confused about its objective. In particular it was not clear whether it was addressing a pre-hospital population, or a severe TBI/ICU population (in which you can measure ICP).

Author response: We are extremely sorry to have caused this confusion. We have tried to alleviate this by changing the title to one that is more appropriate. We have also amended the abstract and throughout to correct this confusion.   

Comment: In the PICO question:

-The population includes patients without a cervical collar?

-Primary outcome is “the effects of placement of the cervical collar in the patient with TBI” – this is very non-specific and then is not reported on anyway. Even if it was, I imagine you would need extremely large numbers to identify any associations between collar use and secondary TBI severity

-Secondary outcome is “changes in ICP after cervical collar placement”. By definition this can only be measured with an invasive device, so by definition these are post-op/ICU patients and the impact on ICP in the pre-hospital setting remains unknown. Additionally, this is not a patient centred outcome – yes it’s true that poorly controlled ICP correlates with poorer functional outcomes but is a difference of 4.5mmHg relevant?

Author response: The PICO statement has been clarified as is the pre-hospital versus post-op/ICU. The reviewer is right, ICP is only being measured in the ICU but the application of the collar has been introduced in the prehospital setting. However, we are not referring to prehospital changes in ICP.

Comment: There are a number of unsupported statements in the conclusion:

 “ICP is a contraindication of cervical collars” line 264 

Author response: Deleted.

Comment: “Early removal and evaluation ensured minimal impact on the patient outcome.” Line 279

Author response: Deleted.

Comment: Line 278 says “Emergency Medical Service should strive to remove cervical collars early” but then line 282 seems to support a radiological evaluation which can’t happen pre-hospital.

Author response: Sentences amended to clarify. (now line 390-395).

Comment: Line 308 All of the included studies were also limited to a single type of rigid cervical collar, the Stifneck collar, 309 which is widely used in the UK. – but then you have commented on the german study which used alternative collars.

Author response: Sentence now amended to remove reference to a single type of collar. The addition of the German study is now justified and included in the meta-analysis as recommended by reviewer 1.

Comment: You have not kept to your own inclusion criteria by including data from studies of unassessable quality, foreign language and inadequate study design.

Author response: True but we have done this since there are only 3 studies and just 59 patients that met our inclusion/exclusion criteria. However, adding the other studies allows us to compare ICP measurements only in the narrative sense. You will see that for the meta-analysis, we have used only the studies that are graded moderate quality above.

Comment: At best, I would re-work this as a study demonstrating the small amount of collated data suggesting collars do increase ICP, and highlighting to providers that it may be worthwhile utilising alternative methods of spinal immobilisation (such as tape and sandbags) early in patients with TBI.

Author response: We agree and hence we hope that the revised version better clarifies the meaning of our results.

Comment: There are typos in line 249 and 252 "sue" not "use"

Author response: Thank you. They have now been amended (now Line 325 and 328).

Round 2

Reviewer 1 Report

All comments addressed. Great paper. 

Author Response

Comment: All comments addressed. Great paper.

Response: We are glad that we were able to answer all of your queries. Thank you for your kind comment.

Reviewer 2 Report

Thanks for the response.

I still do not feel this article is ready for publication.
As suggested in the previous feedback, there is an attempt being made to extrapolate the ICP impact of collar placement in neurosurgical ICU patients with ICP monitors (ie post-op, intubated) to TBI patients in the pre-hospital setting. In the included studies, collars were placed for a very short period of time (5-20mins) compared to the pre-hospital and ED use of collars.

As far as I am concerned these are very different populations, using collars in different situations, and I don’t think it is relevant to be commenting on prehospital collar use.

Examples include Line 247 “All of the studies were prospective studies with the primary common theme that prehospital application of cervical collars in patients with TBI increased ICP during collar application.” This is just not true, none of the studies measured the impact of the collar on ICP pre-hospitally.

Line 251 “Our findings indicate that most of the studies disagree that cervical collars should be used for patients with TBI in the prehospital setting” if they do then it is solely based on author opinion, as none of their methods examined this cohort of patients.

We will never be able to measure ICP in pre-hospital patients. As suggested previously, I think all you can do with this data is say collars probably put the ICP up, so clinicians should aim for early removal in severe TBI in favour of alternative immobilisation methods.

Author Response

Comment 1: I still do not feel this article is ready for publication. As suggested in the previous feedback, there is an attempt being made to extrapolate the ICP impact of collar placement in neurosurgical ICU patients with ICP monitors (ie post-op, intubated) to TBI patients in the pre-hospital setting. In the included studies, collars were placed for a very short period of time (5-20mins) compared to the pre-hospital and ED use of collars. As far as I am concerned these are very different populations, using collars in different situations, and I don’t think it is relevant to be commenting on prehospital collar use.

Author response: We agree with the reviewer and we are sorry that in the revised version we failed to get this across. However, we have been through the whole manuscript and have amended sections that relate to prehospital use of collars, as none of the studies measured ICP in the prehospital setting. Please see clarifications in the Abstract (Lines 17-21), Results (Lines 170-172 and Line 216), Discussion (Lines 253-260; Lines 321-323; Lines 330-333), Conclusions (Line 350).

Comment 2: Examples include Line 247 “All of the studies were prospective studies with the primary common theme that prehospital application of cervical collars in patients with TBI increased ICP during collar application.” This is just not true, none of the studies measured the impact of the collar on ICP pre-hospitally.

Author response: We agree that ICP cannot be measured pre-hospitally and so we cannot say anything about pre-hospital application of collars. Therefore, we have amended this sentence to read: “All of the studies were prospective studies with the primary common theme that TBI patients arrived at the hospital wearing collars. On re-application of the collar during the testing period there was an increase in ICP, which was statistically significant based on our meta-analysis” (now Lines 253-256)

Comment 3: Line 251 “Our findings indicate that most of the studies disagree that cervical collars should be used for patients with TBI in the prehospital setting” if they do then it is solely based on author opinion, as none of their methods examined this cohort of patients.

Author response: We have amended the sentence to read: “Our findings indicate that cervical collar application significantly raises the ICP in TBI patients and hence clinicians should aim for early removal of collars n TBI patients in favor of alternative immobilization methods” (now Lines 257-260)

Comment 4: We will never be able to measure ICP in pre-hospital patients. As suggested previously, I think all you can do with this data is say collars probably put the ICP up, so clinicians should aim for early removal in severe TBI in favour of alternative immobilisation methods.

Author response: We agree and hence we have amended the sections as referred to in response to Comment 1 above.

Round 3

Reviewer 2 Report

This revision is good as it no longer links the pre-hospital events to the ICP changes measured with collar re-application in ICU.

There are a few grammatical/typo errors in the revies text

Line 20

Line 22

Line 167

Line 254

Author Response

Comment: This revision is good as it no longer links the pre-hospital events to the ICP changes measured with collar re-application in ICU.

Author response: Thank you for your time and feedback. Much appreciated.

Comment: There are a few grammatical/typo errors in the revised text: Line 20, Line 22, Line 167, Line 254.

Author response: typos and sentence in line 254 amended.